# Intracellular Antibodies for Drug Discovery and as Drugs of the Future

**DOI:** 10.3390/antib12010024

**Published:** 2023-03-16

**Authors:** T. H. Rabbitts

**Affiliations:** Institute of Cancer Research, Division of Cancer Therapeutics, 15 Cotswold Road, Sutton, London SM2 5NG, UK; terry.rabbitts@icr.ac.uk

**Keywords:** intracellular antibodies, macrodrugs, domain antibodies, iDAbs, biodegraders, delivery, therapy, warheads, PROTAC

## Abstract

The application of antibodies in cells was first shown in the early 1990s, and subsequently, the field of intracellular antibodies has expanded to encompass antibody fragments and their use in target validation and as engineered molecules that can be fused to moieties (referred to as warheads) to replace the Fc effector region of a whole immunoglobulin to elicit intracellular responses, such as cell death pathways or protein degradation. These various forms of intracellular antibodies have largely been used as research tools to investigate function within cells by perturbing protein activity. New applications of such molecules are on the horizon, namely their use as drugs *per se* and as templates for small-molecule drug discovery. The former is a potential new pharmacology that could harness the power and flexibility of molecular biology to generate new classes of drugs (herein referred to as macrodrugs when used in the context of disease control). Delivery of engineered intracellular antibodies, and other antigen-binding macromolecules formats, into cells to produce a therapeutic effect could be applied to any therapeutic area where regulation, degradation or other kinds of manipulation of target proteins can produce a therapeutic effect. Further, employing single-domain antibody fragments as competitors in small-molecule screening has been shown to enable identification of drug hits from diverse chemical libraries. Compounds selected in this way can mimic the effects of the intracellular antibodies that have been used for target validation. The capability of intracellular antibodies to discriminate between closely related proteins lends a new dimension to drug screening and drug development.

## 1. Introduction

Intracellular antibodies are simply antibodies, or fragments of antibodies, that are artificially expressed or delivered to the inside cells, where they function by interacting with target antigens through the antibody-complementarity-determining regions (CDRs) rather than their normal situation of circulating in the blood stream to interact with antigens in serum or on the surface of cells or viruses. Full immunoglobulins such as IgG antibodies comprise two heavy (H) and two light (L) chains linked by covalent disulphide bonds, as illustrated in Figure 1A,B. The original intracellular antibodies were shown by expression of H and L chains in yeast cells [1] and in transfected mammalian cells [2]. While the former studies showed both the H and L chain (a lambda L chain) were expressed, the disulphide-bonded H-L was not conclusively demonstrated and the influence of the reduced state of the cytoplasm was not fully established. The part of the antibody that binds the antigen epitope is H-chain and L-chain variable (V) regions, and in turn the interacting amino acids in the V region are the paratope. Later work on intracellular antibodies dispensed with the Fc region (which would not be used inside cells), using antibody fragments in the form of single-chain Fv (scFv) comprising H-chain and L-chain variable (V) regions held in a single polypeptide chain (Figure 1A), and was conclusive in showing that the single intracellular antibody fragment could fold inside cells and interact with cognate antigens [3]. This was seen despite the presence of conserved cysteines for intra-chain disulphide bonds. Further, intracellular antibody fragments lack the immunoglobulin Fc region and intracellular antibodies are designed to functionally bind to antigens inside cells; the absence of Fc region is not important. Further, the potential for adding a warhead in place of the Fc, warhead being a generic term for an additional moiety, as indicated in Figure 2) fused to the variable region in place of the Fc) adds major flexibility to intracellular antibody engineering. It was later shown by structural analysis of VH and VL that these S-S bonds are not required for folding *per se* since mutation of the cysteines led to a very small difference in the structures of the V regions [4]. It was shown that single domains are the smallest part of the antibody able to bind antigen (Dabs) [5], and the discovery of camelid antibodies, which have only heavy chains, encouraged the use of single domains as intracellular antibody fragments. Thus, more recently, intracellular antibodies have been reduced to single domains, for instance, human iDAbs [6,7] (Figure 1B) or camelid single-domain VHH (nanobodies) [8,9] or shark-derived V_NAR_ H chains [10]. In this article, I concentrate on human intracellular domain antibody fragments (referred to as iDAbs, as depicted in Figure 1B,C). Many of the same experimental options can be applied to other formats, such as monobodies [11], DARPins [12] and affimers [13], for in-cell use.

## 2. Intracellular Antibody Fragments

Collectively, human iDAbs or VHH nanobodies are a good choice for intracellular antibody use since they are single domains with one paratope comprising critical amino acids from three CDRs. This single paratope simplifies affinity manipulation if required [14], and does not involve linker segments such as those used between VH and VL in scFv format. In diverse scFv libraries, the association between VH and VL is often random to accommodate the hydrophobic interactions that naturally occur [15], indicating that many selected scFvs are only (or predominantly) binding through one of the variable region domains. For instance, an anti-RAS scFv was studied where only the VH showed detectable antigen binding [6]. Single-domain VH and VHH are effective antigen binders and thus lack constraints sometimes incurred by the presence of the VL. By sequencing many iDAbs isolated against a variety of antigens (including LMO2, RAS, CRAF, HOXA9, CMYC, TP53) a consensus framework VH sequence was obtained [7] (depicted in Figure 1C, illustrating the framework of the iDAb and the external CDR loops). The structure of an anti-RAS iDAb with HRAS (Figure 1D) formally validated mutant RAS PPI with signal transduction effector molecules as a cancer target [16]. 

The advantage of using a single domain as an intracellular antibody is at least two fold: (i) affinity manipulation only requires changes in the three CDRs of the iDAb and complementarity with VL is not an issue; (ii) the structure of the iDAb has few constraints on the length of CDRs, and in particular, we have been able to extend the length of CDR3 in our human consensus VH iDAb by up to 25 amino acids without loss of expression levels. This enables interactions of this CDR with pockets and otherwise cryptic regions of an antigen that might otherwise not be recognised as epitopes. An additional practical property that iDAbs have is their use to facilitate production of soluble proteins with otherwise disordered proteins that elude recombinant protein production. Just as had been found by co-expressing the AML1 Runt domain and CBFb in *E. coli* to make a soluble heterodimer [17], soluble LMO2 was expressed when co-expressed with a VH iDAb. This led to the crystal structure of LMO2 and to a model for the formation of the LMO2-multiprotein complex [18]. iDAbs and VHH are thus useful accessories for structural biology analysis.

## 3. Specialising Intracellular Antibodies by Fusing them with Moieties to Affect Cell Phenotype or Viability

The ability of intracellular antibodies to produce a recognisable effect on target cells depends on a number of factors that will vary according to the function of the target protein. The effectiveness of an intracellular antibody is related to the half-life of antibody survival, which obviously reflects duration of contact with the antigen. Potent intracellular antibody inhibitors of protein–protein interaction (PPI) can be achieved, which allows for the blockade of a natural PPI since the Kd of the intracellular antibody can be made in the pM range, which is generally higher than that of the natural PPI. Occupancy is the major factor and slow k_off_ facilitates the PPI inhibitor effect of intracellular antibodies via the prolongation of dimer interaction. 

Various other modifications in intracellular antibodies can be made that render the intracellular antibody more potent. These are referred to as warheads, (summarised in Figure 2), and simple protein engineering can derive new bivalent molecular structures that have the dual function of targeting an intracellular antigen and bringing it into the jurisdiction of an existing cellular process. Among the most potent of these highjacked mechanisms is the relocation of proteins within the cell by appending an endoplasmic reticulum (ER) signal peptide (KDEL, [19]), which locks up target antigens in the ER (Figure 2D). Alternatively, cytoplasmic proteins can be made into nuclear ones by the interacting intracellular antibodies, which have a nuclear localisation signal [20] (NLS, Figure 2C).

Exploiting natural pathways within cells to induce a desired phenotype following binding of intracellular antibodies to their target is a powerful way to bring a target protein under the control of cellular elements not normally involved in this process. Two are illustrated in Figure 2. As a proof of concept, an anti-β-galactosidase scFv was directly fused to procaspase 3 (CP3) to form a dimer of dimers of the scFv-CP3 in contact with tetrameric b-galactosidase protein and inducing apoptosis as a result [21] (Figure 2E). This method (called antibody–antigen interaction-dependent apoptosis (**AIDA**)) was later performed using two separate iDAbs (one VH and one VL) that were cloned from an anti-RAS scFv [22]. The concept for AIDA technology was originally aimed to target fusion proteins that arise from commonly occurring chromosomal translocations, such as BCR-ABL fusion in Philadelphia-positive CML as an exemplar of this common type of tumour-associated protein [23]. In such a scenario, one iDAb-CP3 fusion would bind an epitope on one fusion partner (such as BCR) and a second iDAb-pCP3 would bind to an epitope on the other fusion partner (such as ABL). This generic approach could be applied against the plethora of tumour-associated fusion proteins, with suitable controls to mitigate off-target effects.

Protein degradation was suggested as another route to control cellular phenotype by invoking proteosome degradation of targets in yeast [24]. This was followed by SIT technology in which the proteasome machinery was recruited for the targeted degradation of cellular proteins [25]. It was thus proposed that direct fusion of intracellular antibodies to ubiquitin ligases would cause specific degradation via the proteosome of the target protein after ubiquitination. Iterations of this approach have been developed in which direct fusion of various E3 ligases with intracellular antibody fragments (biodegraders) creates a binary complex in cells to lead to ubiquitination of target proteins and their proteosome degradation (Figure 2F). This is akin to the chemical method called proteolysis-targeting chimaera (PROTAC), in which a ternary complex is created comprising the target, bound by a specific compound linked to an E3 ligase ligand, and an E3 ligase. This results in proteosome degradation of targeted proteins [26], as intracellular antibody-E3 ligase fusions cause binary interactions between two molecules and are less prone to Hook effects. 

Protein degradation using intracellular antibody-E3 ligase fusions can take advantage of the ease with which intracellular antibodies can distinguish between family members and isoforms. The LMO LIM-domain-only family comprises four paralogues (LMO1, 2, 3 and 4), and an anti-LMO2 iDAb has been developed that discriminates against the other paralogues [27]. Similarly, RAS isoforms have been distinguished by a DARPin that selectively binds to the allosteric lobe of KRAS compared to an iDAb that binds the effector domains of H, K and NRAS [28]. In turn, when made into biodegraders, all RAS isoforms or just the KRAS isoform is subjected to protein degradation by the iDAb or DARPin-E3 ligase fusion, respectively [29]). The potential for this extends even further since RAS-mutant-specific macromolecules have been identified as monobodies [11,30], and this informs the possibility of applying technologies to build panels of mutant-protein-specific intracellularly effective macromolecules. 

**Figure 1 antibodies-12-00024-f001:**
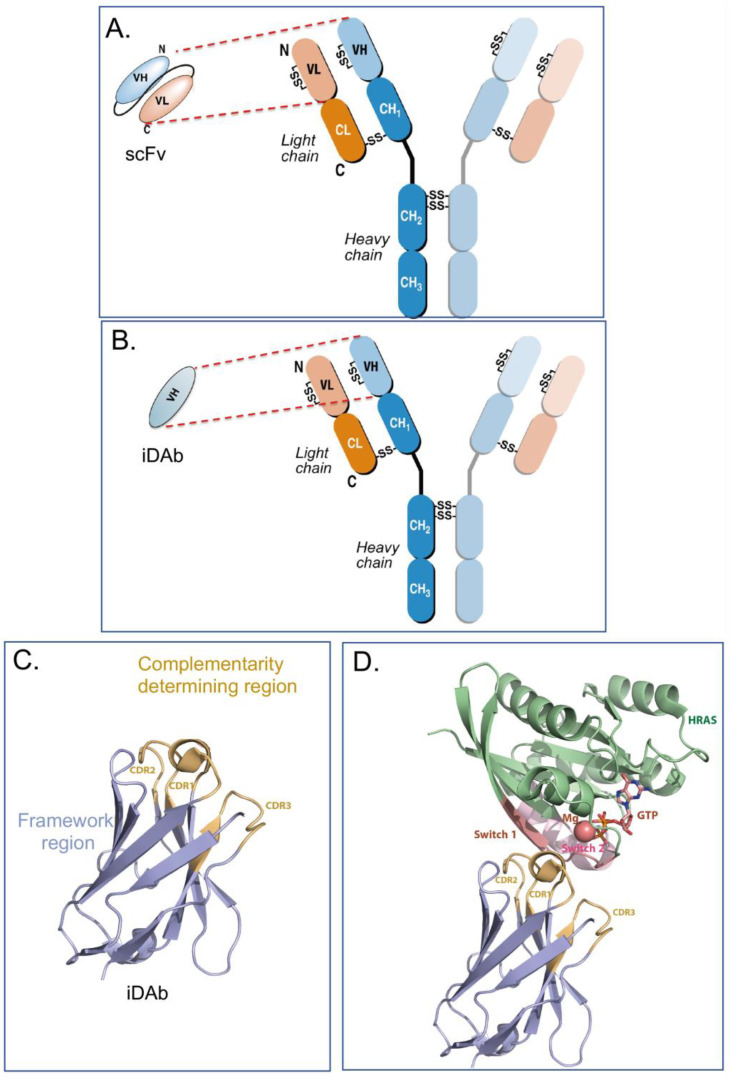
Antibody, intracellular antibody fragments and structure of a consensus human intracellular domain antibody. Immunoglobulins comprise two H and two L chains held together by inter-chain disulphide bonds (panel (**A**,**B**)). The H chain has a V region and different C-region domains (panels (**A**,**B**) show subclass IgG1 as CH1, 2 and 3 with a hinge region between CH1 and CH2). An intra-chain disulphide bond is found in each V region. Antibody fragments used for intracellular antibodies are either scFv (shown in panel (**A**)), where the V regions of H and L chains are held in a single polypeptide chain by a short linker between VH and VL), or a domain antibody, most often VH only (panel (**B**)). Using intracellular antibody selections based on intracellular antibody capture in yeast [7], a consensus VH sequence was derived by comparing the amino acids at each position in the framework residues of several human iDAbs. Panel (**C**) shows the crystal structure of an iDAb VH that binds to the RAS isoforms and the crystal structure of this VH in contact with HRAS [20]. Panel (**D**) shows crystal structure of GTP-bound HRAS interacting with a VH iDAb [20]. For the VH, the framework region is coloured in blue and the CDR regions in brown. For HRAS, the apoenzyme is shown in green and the effector binding (switch) region in purple. The Mg atom and GTP are indicated in HRAS.

## 4. Options for Delivery of Intracellular Antibodies to Cells

The importance of intracellular antibodies derives from their natural properties and precise specificity, indicating their potential as highly selective drugs. At present, intracellular antibodies and similar reagents are potent and versatile laboratory tools used via in vitro transfections or by viral infection. Nonetheless, protein engineering is a flexible approach; it promises a potential and convenient new drug development pipeline with intracellular antibodies. The systemic delivery of intracellular antibodies (or similar modalities, such as DARPins [31] or monobodies [32]), is a main challenge in this field. When this is achieved, it will transform drug development because selected antibody fragments will be able to be optimised via affinity maturation or dematuration as required [33,34] and/or warheads easily engineered. In this context, macromolecules are macrodrugs [35] and potentiate a new pharmacology.

The delivery challenge requires macrodrugs to find their way to the disease cells, enter the cell in sufficient amounts to produce a phenotype and have a lasting effect in the treatment of disease. Several options are being developed to enable the use of intracellular antibodies as macrodrugs. For example, a cell-penetrating IgG anti-RAS antibody has been developed [36], but generally proteins *per se* do not cross the plasma membrane unless cross-linked to surface antigens in the normal process of immune recognition. Cell-penetrating peptides, developed as Antennapedia [37] and HIV TAT-derived sequences [38,39], show great promise in new cyclic formats when used in conjunction with antibodies [40]. The drawbacks of cell-permeable intracellular antibody protein delivery include the quantity of functional antibody that reaches the cytosol, targeting the cells of interest to avoid (if possible) normal cells and the duration of interaction of the antibody with the target before normal cellular protein turnover removes the antibody. While immunogenicity issues in antibody fragments or other intracellular formats can be prevented if delivery to cells is undertaken via the expression route, possible challenges in immunogenicity may be encountered when using systemically delivered protein, such as that coupled to CPPs. However, work with camelid nanobody VHH or with shark V_NAR_ H chains has shown their minimal immunogenicity [41]. 

The alternative for systemic delivery of macrodrugs is to use nucleic acid cargoes, encoding intracellular antibodies, in vehicles. This new approach has gained credence since the application of either mRNA or viral genomic DNA in SARS-CoV-2 vaccination programmes [42]. In these applications, vehicles such as lipid nanoparticles (LNPs), with encapsulated macrodrug mRNA, or as viruses, such as adenovirus or adeno-associated virus, with macrodrug genes could deliver the nucleic acids for macrodrug expression in targeted cells. A potential advantage of these approaches is that a more sustained amount of macrodrug would be produced from the in-cell expression, but familiar drawbacks relate to the amount of either mRNA released from the LNPs or the genes encoded by the infecting virus particles and the consequential scale of protein production [43]. 

A potential way to enhance cell selectivity and magnitude of uptake is to coat the delivery vehicles with target-cell-selective surface ligands that will interact with disease-cell expressed surface (CD) markers. These ligands can be antibody fragments that recognise the target cell surface protein or a smaller ligand for the marker. Attempts to define such tumour-specific surface proteins have utilised high-density RNA-seq data to compare tumour cells with normal counterparts, e.g., by employing a surfaceome database created from known gene information [44]. With this strategy, two situations were examined where the probable cancer cell of origin was known (i.e., Ewing sarcoma [44] and T-cell leukaemia [45]). Alternatively, proteomic analysis can potentially guide the identification of CD proteins of interest if a sufficient depth of analysis can be achieved. Finally, it may be that dual targeting will be beneficial in achieving better target cell specificity [46] and increasing the valency of antibodies using multimerisation domains; for example, the p53 tetramerisation domain [47,48] could add potency to monospecific and bispecific antibodies. The question remains, however, whether this vehicle tagging strategy will enhance the biodistribution of systemically delivery vehicles or whether the real effect will be to enhance local uptake of vehicles into target cells in preference to normal cells, when the tagged vehicles reach the tumour mass [49]. 

An alternative to using intracellular antibodies as deliverable macrodrugs could be applied to disease via genome editing with intracellular antibody genes in target cells using CRISPR/cas9 methods [50]. In this concept, an anti-viral protein biodegrader gene could potentially be introduced into haematopoietic stem cell genomes to become specifically expressed in restricted progeny cells. With advanced technology developments, this could be applied, for instance, for HIV in AIDS within economically limited populations and for EBV in Burkitt’s lymphoma in the malarial endemic belt.

## 5. Intracellular-Antibody-Derived Compounds: Bridging the Gap between Antibodies and Small Molecules

The difficulties in developing reliable and general delivery methods for using intracellular antibodies as macrodrugs *per se* prompted the development of Antibody-derived compound technology (**Abd** methodology) [51]. This was based on the observation that the iDAb CDR (the paratope) binding the antigen epitope has an approximately ten-amino-acid footprint and, based on molecular analysis [52], this would predict the small molecular equivalent of about 500 dA, just at the limit of the Lipinsky Rule of 5 which was an empirical evaluation of how drug-like a compound would be [53]. Accordingly, we tested this possibility with the so-called undruggable RAS proteins. In the first-generation exposition of this **Abd** approach, a chemical fragment library was screened with HRAS protein, and chemical hits at the paratope–epitope interface were identified via competition with the intracellular antibody [51,54]. The key factor in this success was the pM K_d_ of the iDAb-RAS interaction (with low k_off_), which maintained the interaction of antigen–antibody during competition assessment in surface plasmon resonance. The intracellular antibody was in this case inhibiting compounds binding to the target.

Second-generation **Abd** was carried out on KRAS using a lower-affinity antibody fragment in order to find compounds that would directly inhibit the paratope–epitope interaction in the screen. As this requires low-affinity paratope–epitope interaction, a simple iDAb dematuration protocol was developed that depends only on the knowledge of the primary sequence of the iDAb [33] (i.e., no structural data are needed). These methods produced chemical matter that was pan-RAS since the iDAb involved binding to the RAS effector interaction region. The compounds were PPI inhibitors, as shown by cell-based BRET biosensors [55]. 

The third-generation **Abd** technology was designed to find compounds that displace intracellular antibodies that bind to disordered proteins, among which are many chromosomal translocation proteins and transcription factors. This disordered protein Abd screen used an anti-LMO2 intracellular antibody that had been used to confirm target validation in preclinical models [27]. It also implemented the dematured intracellular antibody approach but in a cell-based BRET screen, where the disordered protein was expressed in its normal cellular environment. Thus, this allowed chemical library screening of a disordered protein [34] that could not be expressed well in recombinant form without co-expression of the iDAb [18]. This cell-based **Abd** methodology was exemplified by work on the LMO2 T-cell oncogenic protein, which was discovered to be a chromosomal translocation-activated gene in T-ALL [56,57] (reviewed in [58]). LMO2 is transcription factor [59] and, like other transcription factors, was considered hard-to-drug. The methods that led to **Abd** LMO2-binding compounds provide a generic route to similar compounds for other transcription factors and disordered proteins, from target validation to drug discovery.

The **Abd** technology is an antibody-based approach for drug discovery. It can not only be applied to intracellular antibodies but also to antibodies against the spectrum of diseases such as COVID-19, HIV and Ebola. An antibody binding to the membrane proximal external region (MPER) of the HIV-1 envelope has been used to guide the selection of small molecules that may be developed into therapeutic alternatives to the antibody [60]. The antibody-based approach to compound identification may be applicable to other clinical indications to replace antibodies where the cost of goods is very high and, without half-life extension properties, can be deleterious to patients due to the frequency of antibody treatment. Orally available chemical drugs are much more advantageous for patient use and compliance, as well as economy benefit, if specificity can be maintained while potency is increased. In particular, inducing the transition from antibody to small molecule is potentially a rapid route to drug development and may be applied in future cases of pandemics, like we have recently experienced with SARS-CoV-2, where small-molecule, orally available drugs were urgently needed worldwide. For **Abd** compound surrogates of intracellular antibodies, their current advantages are also great because of difficulties in systemically delivering vehicles with macrodrug cargoes.

## 6. Conclusions

Intracellular antibodies, and other antigen-binding macromolecules formats, have at least three de novo uses: first, as research tools for studying cell biology and protein function; second, as templates for drug discovery using the antibody paratope; and third, as potential drugs in their own right. For intracellular antibodies, their natural properties of antigen selectivity, discrimination and high-affinity binding are defining factors. Intracellular antibodies have flexibility and potential for very high potency compared to small molecules, which are limited by the Rule of 5 [53] or internalising macrocycle peptides [61]. They can readily be modified using protein engineering to include moieties that produce effector reactions inside cells but also can be used for target validation prior to the initiation of small-molecule drug development campaigns. The major disadvantage of intracellular antibodies is currently the lack of effective methods for their delivery either as proteins or in the form of expressible nucleic acids.

Intracellular antibodies can be selected against any cellular protein, from nuclear transcription factors to cytoplasmic proteins to plasma-membrane-associated proteins. They can also be engineered to carry warheads that can change their function from merely binding proteins to ones that can induce phenotypic change (Figure 2). iDAbs can bind to the so-called undruggable targets such as RAS and hard-to-drug targets such as transcription factors. Because they have the natural antibody property of specificity, they can be used to discriminate between isoforms and paralogues (e.g., the LMO family of transcription factors [27]) and even mutants of the same protein (e.g., RAS mutants by monobodies [30]). In the case of iDAbs, because they are single domains, paratope manipulation is easily achieved without any 3-D structural data, so increased or decreased affinity can be readily accomplished based on the primary sequence alone.

What could a drug development programme based on intracellular antibodies be like? This is illustrated in Figure 3, starting from a diverse iDAb library, screened using the target protein either via a phage display followed by yeast intracellular capture or only the latter (Figure 3A). Target-specific iDAbs are characterised by the hits, and those that perform the function required (e.g., as PPI inhibitors) are taken to Step 2, where the target validation is undertaken in either cell assays or mouse modelling (using expression vectors to produce the iDAb). The first stages therefore have four phases (Figure 3B) during which valuable data are obtained in the first stages Figure 3A), such as the implementation of iDAbs as tools for exploring the proteome, e.g., dissecting protein function. 

When suitable intracellular antibodies have been identified, there are currently two options for taking these towards drug discovery. The first is using the iDAb-binding surface (the paratope) to select small compounds from chemical libraries using **Abd** technology (Figure 3C). **Abd** compounds are iDAb surrogates and chemical matter that can be used for drug development through hit-to-lead and lead optimisation (Figure 3D). **Abd** technology is an approach that can bring new drug discovery opportunities to the landscape of disordered proteins, and can be used to discriminate members of protein families and even alternative proteins arising via alternative splicing.

An exciting alternative is the direct use of iDAbs (as proteins or nucleic acids for expression) as macrodrugs (Figure 3C). This requires the development of new technology but has huge potential for application in the whole gamut of clinical situations. The choice of macrodrug delivery vehicle (Figure 3C, step 2) will be dictated by the type of disease. In the case of oncology applications, immediate effects mediated by the macrodrug protein itself may be sufficient to trigger cell death. Applications in infections may also be targeted by protein macrodrug cargoes. For application for long-term clinical indications such as neuropathy or inflammation, nucleic acid cargoes are the most attractive option to produce more sustained levels of macrodrug. The future use of armed intracellular antibodies for drugs *per se* is an exciting opportunity for expanding the available drug network into a new pharmacology. 

In the use of intracellular antibodies for drug discovery or use as macrodrugs (here encompassing all the many exciting formats of macromolecular proteins), the ultimate goal is to follow preclinical testing in animal models with first-in-human studies (phase I) followed by more focused clinical trials (Figure 3D). Success in these endeavours will advance intracellular antibodies from the discovery phase of research to their use as new therapeutics.

**Figure 2 antibodies-12-00024-f002:**
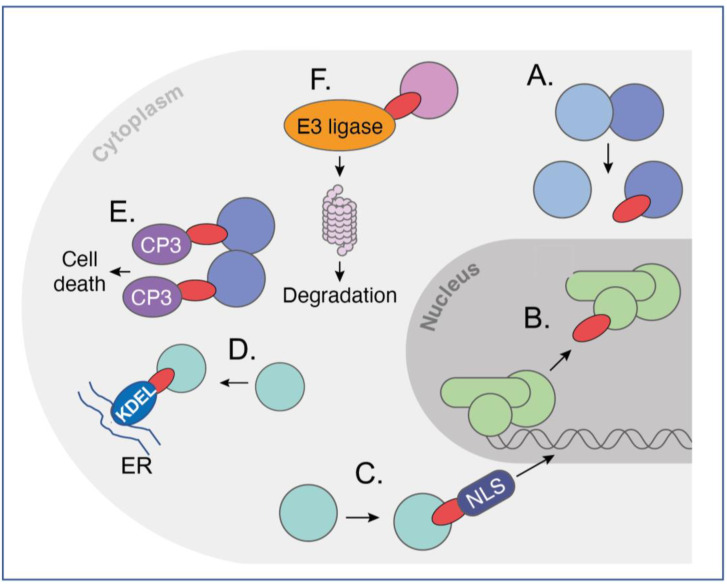
Versatility of intracellular antibodies invoking cellular pathways. Intracellular antibodies can be used in many formats, from whole immunoglobulin to scFv to single variable regions domains. The figure depicts several potential functions of intracellular antibodies. The diagram shows sev-eral, nonexclusive, independent uses of iDAbs (shown here as 

) inside cells that have differ-ent functionalities, ranging from iDAb protein (**A**,**B**) to those fused to effector moieties (warheads) that invoke a cellular property for the effectiveness of the iDAb (**C**–**F**). In (**A**): an iDAb acts as an in-hibitor of a protein–protein interaction; in (**B**): an iDAb acts as an inhibitor of a transcription factor interaction with DNA (this could also be applied to protein-RNA interaction); in (**C**): an iDAb fused to an NLS relocates its protein target from the cytoplasm to nucleus; in (**D**): an iDAb fused with a endoplasmic reticulum signal sequence (KDEL) sequesters a target protein in the endoplasmic re-ticulum; in (**E**): two iDAbs linked to procaspase 3 (CP3) bind to a target protein and cause auto-cleavage of procaspase 3 to active caspase 3, thereby initiating programmed cell death in an anti-gen-dependent fashion, as may occur to target chromosomal translocation fusion proteins; in (**F**): iDAbs are turned into biodegraders to induce targeted protein degradation, by fusing an iDAb to an E3 ligase, allowing a binary complex to form with the target protein, resulting in ubiquitination of the protein and proteasomal degradation. These various functionalities can also be applied with the use of DARPins, monobodies and affimers. This figure is adapted from a previous publication [62]. The figure was created by Claudia Stocker, Vivid Biology.

**Figure 3 antibodies-12-00024-f003:**
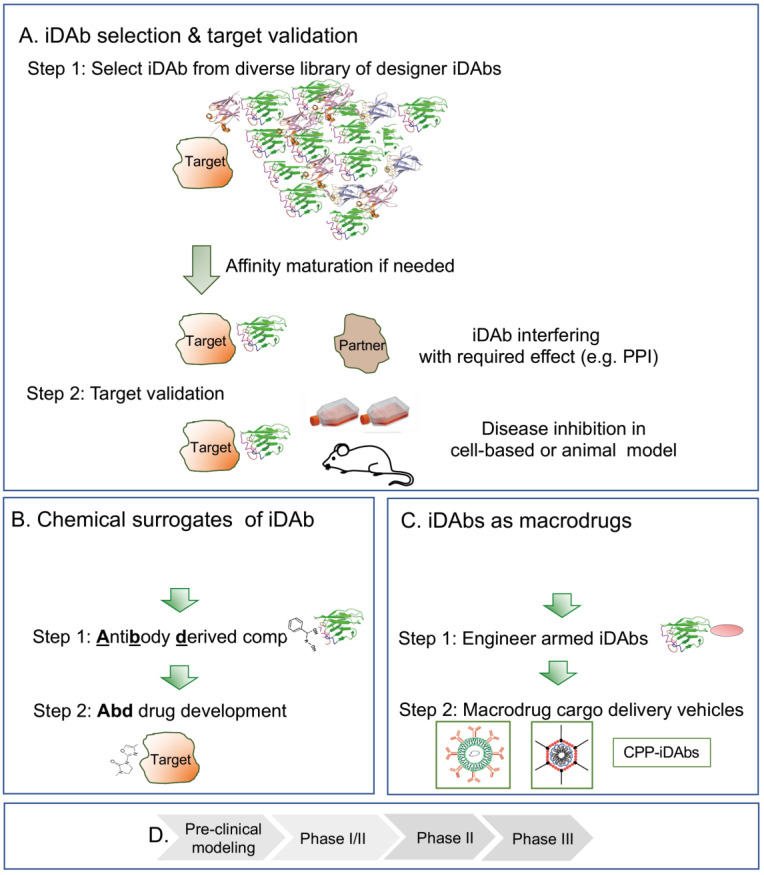
Application of intracellular antibodies for discovery research, target validation and novel therapeutics. Intracellular antibodies are tools for disease target validation (panel (**A**,**B**)) and precursors that can be used in chemical library screens for small-molecule drug development (panel (**C**,**D**)) or as drugs *per se* (herein macrodrugs). Panel (**A**): for target validation step 1, iDAbs that bind to the selected target protein are obtained via screening a library of diverse iDAbs, followed by identification of a suitable iDAb that has the property required (as shown in panel (**A**), line 3, an inhibitor of a protein–protein interaction). In step 2, this iDAb is then used for target validation in a preclinical model of interest (such as a mouse in vivo model or a tissue culture in vitro model, line). When the intracellular antibody tool is confirmed, two options are available for the next step. Panel (**B**) outlines the selection of small-molecule chemical surrogates of the iDAb binding site (the paratope) using competitive small-molecule library screening (step 1) to obtain chemical matter for drug development campaigns (step 2). This is the Antibody-derived compound technology (**Abd** technology). In this approach, the antibody is replaced by a chemical compound for conventional for hit-to-lead optimisation. The second application is the use of the intracellular antibody as a drug in its own right (a macrodrug) (panel (**C**)). Here, an iDAb or engineered iDAb armed with an effector warhead for enhanced performance (step 1) is delivered into cells using nanoparticles with encapsulated mRNA, viral vectors or cell-penetrating peptides, CPPs (step 2). In both the Abd technology (panel (**C**)) and macrodrug delivery (panel (**C**)) scenarios, preclinical testing would follow optimisation, followed by first-in-man phase I/II trials (**D**). This would be the advent of a new pharmacology involving macrodrugs. The strategies illustrated in the figure are shown for iDAbs but could be applied to other protein macromolecules, such as VHHs, V_NARs_, DARPins, monobodies and affimers.

## Data Availability

No new data were created or analyzed in this study. Data sharing is not applicable to this article.

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
