# Peer review of "Intracellular Antibodies for Drug Discovery and as Drugs of the Future"

_2073-4468, 2023, doi:10.3390/antib12010024_

Round 1

Reviewer 1 Report

This is an interesting review of the concept and potential applications of the intracellular produced antibodies. Because this topic is not well familiar to the majority of the readership - more detailed introduction of the concept of the intracellular antibodies and the way they are made needs to be included. In this regard, Figures 2 and 3 which are supposed to be helping to understand the design and the applications of such antibodies need much more detailed legends explaining all the abbreviations etc. The term "warhead" is a scientific jargon  which should be replaced with the proper explanation of what cytotoxic moieties these antibodies will be conjugated to. Finally, a paragraph clearly describing possible advantages of intracellular antibodies over easily internalizing peptides and small molecules should be included.

Reviewer 2 Report

The authors summarize the recent progress of intracellular antibody development. Overall the article is interesting, well-written, and may eventually be worthy of publishing. However, the reviewer feels that a little amendment may strengthen the overall claim. In particular, a clear explanation of the pros/cons between antibody fragments and antibodies should be considered.

1.         The reviewer agrees that one of the advancements of antibody fragments described in this manuscript is rapid internalization compared with antibodies. If the authors summarize the comparison of the internalization rate of both conjugates (antibody fragments: XX% and antibody: YY%), it should strengthen this manuscript. Of course, these rates depend on the behavior of the binder molecule (mAb VS mAb fragment), but rough prediction still is useful

2.         One remaining challenge of antibody fragments is the risk of immunogenicity. The authors should have comments about this topic. To summarize current development of immunogenicity may be useful.

3.         Also potential disadvantage of the antibody fragments is the lack of the Fc portion causing a short half-life (due to lack of FcRn recycling), and a lack of Fc receptor-mediated activities (ADCC, ADCP, etc). The reviewer is not sure the antibody fragment is superior to antibodies even if considering rapid penetration.

4.         The reviewer thinks one of the promising approaches for utilizing intracellular antibodies is an application to conjugates consisting of antibodies and cytotoxic payloads. The authors should have comments on that. Recently some site-specific conjugation can modify antibody fragments to conjugate with cytotoxic payloads such as Bioorg Med Chem Lett. 2021, 128360. The citation of this article may help to show the future direction of this study.

Round 2

Reviewer 1 Report

The author has done an excellent job revising the manuscript which has greatly improved.

Author Response

receive

Reviewer 2 Report

The reviewer understood the author's response.

Author Response

receive